# Rethinking Importance Weighting for Deep Learning under Distribution Shift

**Tongtong Fang**[*1†]   **Nan Lu**[*1,2]   **Gang Niu**[2‡]   **Masashi Sugiyama**[2,1]

[1]The University of Tokyo, Japan       [2]RIKEN, Japan

## Abstract

Under *distribution shift* (DS) where the training data distribution differs from the test one, a powerful technique is *importance weighting* (IW) which handles DS in two separate steps: *weight estimation* (WE) estimates the test-over-training density ratio and *weighted classification* (WC) trains the classifier from weighted training data. However, IW cannot work well on complex data, since WE is incompatible with deep learning. In this paper, we rethink IW and theoretically show it suffers from a *circular dependency*: we need not only WE for WC, but also WC for WE where a trained *deep classifier* is used as the *feature extractor* (FE). To cut off the dependency, we try to pretrain FE from unweighted training data, which leads to biased FE. To overcome the bias, we propose an end-to-end solution *dynamic IW* that iterates between WE and WC and combines them in a seamless manner, and hence our WE can also enjoy deep networks and stochastic optimizers indirectly. Experiments with two representative types of DS on three popular datasets show that our dynamic IW compares favorably with state-of-the-art methods.

## 1   Introduction

Supervised *deep learning* is extremely successful [12], but the success relies highly on the fact that training and test data come from the same distribution. A big challenge in the age of deep learning is *distribution/dataset shift* (DS) [40, 48, 38], where training and test data come from two different distributions: the training data are drawn from $p_{\mathrm{tr}}(\boldsymbol{x}, y)$, the test data are drawn from $p_{\mathrm{te}}(\boldsymbol{x}, y)$, and $p_{\mathrm{tr}}(\boldsymbol{x}, y) \neq p_{\mathrm{te}}(\boldsymbol{x}, y)$. Under DS, supervised deep learning can lead to *deep classifiers* (DC) *biased to the training data* whose performance may significantly drop on the test data.

A common practice is to assume under DS that $p_{\mathrm{te}}(\boldsymbol{x}, y)$ is *absolutely continuous* w.r.t. $p_{\mathrm{tr}}(\boldsymbol{x}, y)$, i.e., $p_{\mathrm{tr}}(\boldsymbol{x}, y) = 0$ implies $p_{\mathrm{te}}(\boldsymbol{x}, y) = 0$. Then, there exists a function $w^*(\boldsymbol{x}, y) = p_{\mathrm{te}}(\boldsymbol{x}, y)/p_{\mathrm{tr}}(\boldsymbol{x}, y)$, such that for any function $f$ of $\boldsymbol{x}$ and $y$, it holds that

$$\mathbb{E}_{p_{\mathrm{te}}(\boldsymbol{x}, y)}[f(\boldsymbol{x}, y)] = \mathbb{E}_{p_{\mathrm{tr}}(\boldsymbol{x}, y)}[w^*(\boldsymbol{x}, y) f(\boldsymbol{x}, y)]. \tag{1}$$

Eq. (1) means after taking proper weights into account, the weighted expectation of $f$ over $p_{\mathrm{tr}}(\boldsymbol{x}, y)$ becomes *unbiased* no matter if $f$ is a *loss* to be minimized or a *reward* to be maximized. Thanks to Eq. (1), *importance weighting* (IW) [46, 49, 21, 50, 51, 25] can handle DS in two separate steps:

- *weight estimation* (WE) with the help of a tiny set of validation data from $p_{\mathrm{te}}(\boldsymbol{x}, y)$ or $p_{\mathrm{te}}(\boldsymbol{x})$;
- *weighted classification* (WC), i.e., classifier training after plugging the WE result into Eq. (1).

IW works very well (e.g., as if there is no DS) if the form of data is simple (e.g., some linear model suffices), and it has been the common practice of non-deep learning under DS [52].

---

[*]Equal contribution.

[†]Preliminary work was done when TF was a master student at KTH and an intern student at RIKEN.

[‡]Correspondence to: GN <gang.niu@riken.jp>, TF <fang@ms.k.u-tokyo.ac.jp>

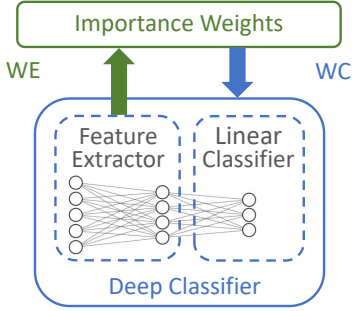

Blue arrow depicts WC depending on WE; green arrow depicts WE depending on WC—this makes a circle.

Figure 1: Circular dependency.

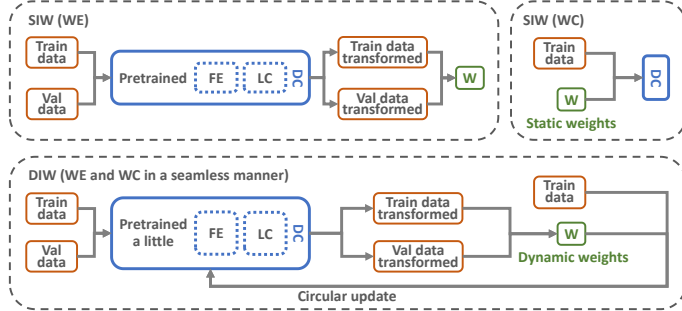

SIW/DIW stands for static/dynamic importance weighting; FE is short for feature extractor, and LC/DC is for linear/deep classifier; W is a set of weights. Circular update is employed to solve circular dependency.

Figure 2: Illustrations of SIW and DIW.

However, IW cannot work well if the form of data is complex [5]. Consider a $k$-class classification problem with an input domain $\mathcal{X} \subset \mathbb{R}^d$ and an output domain $\mathcal{Y} = \{1, \ldots, k\}$ where $d$ is the input dimension, and let $\boldsymbol{f} : \mathcal{X} \to \mathbb{R}^k$ be the classifier to be trained for this problem. Here, $w^*$ processes $(d+1)$-dimensional or $(d+k)$-dimensional input depending on how $y$ is encoded and $\boldsymbol{f}$ processes $d$-dimensional input, and consequently WE is not necessarily easier than WC. Hence, when a deep model is needed in WC, more *expressive power* is definitely needed also in WE.

In this paper, we improve IW for deep learning under DS. Nevertheless, WE and WC are different tasks with different goals, and it is difficult to boost the expressive power of WE for three reasons:

- some WE methods are *model-free*, i.e., they assign weights to data without a model of $w^*$;
- other WE methods are *model-based* and also *model-independent*, but the optimizations are constrained due to $\mathbb{E}_{p_{\mathrm{tr}}(\boldsymbol{x},y)}[w^*(\boldsymbol{x},y)] = \mathbb{E}_{p_{\mathrm{te}}(\boldsymbol{x},y)}[1] = 1$ and incompatible with stochastic solvers;
- most powerful deep models nowadays are hard to train with the WE optimizations since they are *designed for classification*, even if we ignore the constraint or satisfy it within each mini-batch.

Therefore, it sounds better to boost the expressive power by an external *feature extractor* (FE). For instance, we may rely on $\boldsymbol{f}$ that is a deep model chosen for the classification problem to be solved. Going along this way, we encounter the *circular dependency* in Figure 1: originally we need $w^*$ to train $\boldsymbol{f}$; now we need a trained $\boldsymbol{f}$ to estimate $w^*$. It becomes a chicken-or-egg causality dilemma.

We think of two possible ways to solve the circular dependency, one *pipelined* and one *end-to-end*. The pipelined solution pretrains a DC from unweighted training data, and creates the FE from this DC; then, WE is done on the data transformed by the FE. Since the weights cannot change, we call this method *static importance weighting* (SIW), as illustrated in the top diagram of Figure 2. Here, the DC is biased to the training data, and so is the FE, which could be empirically confirmed. As a result, this naive pipelined solution is only slightly better than no FE unfortunately.

To overcome the bias of SIW, we propose *dynamic importance weighting* (DIW) as the end-to-end solution; see the bottom diagram of Figure 2. DIW iterates between WE (on the transformed data) and WC (for updating the DC and FE) and combines them in a seamless manner. More specifically, let $\mathcal{W}$ be the set of importance weights initialized to be all ones, and let $\boldsymbol{f}$ be initialized randomly. Subsequently, we update $\boldsymbol{f}$ for several epochs to pretrain it a little, and then we update both $\mathcal{W}$ and $\boldsymbol{f}$ for the remaining epochs. In each mini-batch, $\mathcal{W}$ is computed by the objective of WE (we adopt *kernel mean matching* [21] in our DIW implementation) where $\boldsymbol{f}$ is fixed, and then $\boldsymbol{f}$ is updated by the objective of WC where $\mathcal{W}$ is fixed in backpropagation.[1] As a consequence, this more advanced end-to-end solution can gradually *improve* $\mathcal{W}$ and *reduce the bias of* $\boldsymbol{f}$, which suggests that IW for deep learning nowadays can work as well as IW for non-deep learning in the old days hopefully.

The rest of the paper is organized as follows. DIW is proposed in Sec. 2 with its applications given in Sec. 3. The related research topics for handling DS are discussed in Sec. 4. The experiments are presented in Sec. 5. Some more experimental results can be found in the appendices.

## 2 Dynamic importance weighting

As mentioned earlier, under distribution shift, training and test data come from two different distributions $p_{\mathrm{tr}}(\boldsymbol{x}, y)$ and $p_{\mathrm{te}}(\boldsymbol{x}, y)$ [40, 48]. Let $\{(\boldsymbol{x}_i^{\mathrm{tr}}, y_i^{\mathrm{tr}})\}_{i=1}^{n_{\mathrm{tr}}}$ be a set of i.i.d. training data sampled from $p_{\mathrm{tr}}(\boldsymbol{x}, y)$ where $n_{\mathrm{tr}}$ is the training sample size, and $\{(\boldsymbol{x}_i^{\mathrm{v}}, y_i^{\mathrm{v}})\}_{i=1}^{n_{\mathrm{v}}}$ be a set of i.i.d. validation data sampled from $p_{\mathrm{te}}(\boldsymbol{x}, y)$ where $n_{\mathrm{v}}$ is the validation sample size. We assume validation data are much less than training data, namely $n_{\mathrm{v}} \ll n_{\mathrm{tr}}$, otherwise we can use validation data for training.

**Weighted classification** From now on, we assume our classifier $\boldsymbol{f}$ to be trained is a deep network parameterized by $\theta$, denoted by $\boldsymbol{f}_\theta$. Let $\ell : \mathbb{R}^k \times \mathcal{Y} \to \mathbb{R}_+$ be a *surrogate loss function* for $k$-class classification, e.g., *softmax cross-entropy loss*. The classification risk of $\boldsymbol{f}_\theta$ is defined as

$$R(\boldsymbol{f}_\theta) = \mathbb{E}_{p_{\mathrm{te}}(\boldsymbol{x}, y)}[\ell(\boldsymbol{f}_\theta(\boldsymbol{x}), y)], \tag{2}$$

which is the performance measure we would like to optimize. According to Eq. (1), if $w^*(\boldsymbol{x}, y)$ is given or $\mathcal{W}^* = \{w_i^* = w^*(\boldsymbol{x}_i^{\mathrm{tr}}, y_i^{\mathrm{tr}})\}_{i=1}^{n_{\mathrm{tr}}}$ is given, $R(\boldsymbol{f}_\theta)$ can be approximated by

$$\widehat{R}(\boldsymbol{f}_\theta) = \tfrac{1}{n_{\mathrm{tr}}} \sum_{i=1}^{n_{\mathrm{tr}}} w_i^* \ell(\boldsymbol{f}_\theta(\boldsymbol{x}_i^{\mathrm{tr}}), y_i^{\mathrm{tr}}), \tag{3}$$

which is the objective of WC. With the *optimal weights*, the weighted empirical risk in Eq. (3) is an *unbiased estimator* of the risk in Eq. (2), and hence the trained classifier as the minimizer of $\widehat{R}(\boldsymbol{f}_\theta)$ should converge to the minimizer of $R(\boldsymbol{f}_\theta)$ as $n_{\mathrm{tr}}$ approaches infinity [46, 49, 21, 50, 51, 25].

**Non-linear transformation of data** Now, the issue is how to estimate the function $w^*$ or the set $\mathcal{W}^*$. As discussed earlier, we should boost the expressive power externally but not internally. This means we should apply a *non-linear transformation* of data rather than directly model $w^*(\boldsymbol{x}, y)$ or $p_{\mathrm{tr}}(\boldsymbol{x}, y)$ and $p_{\mathrm{te}}(\boldsymbol{x}, y)$ by deep networks. Let $\pi : \mathcal{X} \times \mathcal{Y} \to \mathbb{R}^{d_{\mathrm{r}}}$ or $\pi : \mathcal{X} \times \mathcal{Y} \to \mathbb{R}^{d_{\mathrm{r}}-1} \times \mathcal{Y}$ be a transformation where $d_{\mathrm{r}}$ is the reduced dimension and $d_{\mathrm{r}} \ll d$; let $\boldsymbol{z} = \pi(\boldsymbol{x}, y)$ be the transformed random variable, whose source of randomness is $(\boldsymbol{x}, y)$ exclusively. By applying $\pi$, we expect that WE on $\boldsymbol{z}$ will be much easier than WE on $(\boldsymbol{x}, y)$. The feasibility of applying $\pi$ is justified below.

**Theorem 1.** *For a fixed, deterministic and invertible transformation $\pi : (\boldsymbol{x}, y) \mapsto \boldsymbol{z}$, let $p_{\mathrm{tr}}(\boldsymbol{z})$ and $p_{\mathrm{te}}(\boldsymbol{z})$ be the probability density functions (PDFs) induced by $p_{\mathrm{tr}}(\boldsymbol{x}, y)$, $p_{\mathrm{te}}(\boldsymbol{x}, y)$, and $\pi$. Then,*

$$w^*(\boldsymbol{x}, y) = \frac{p_{\mathrm{te}}(\boldsymbol{x}, y)}{p_{\mathrm{tr}}(\boldsymbol{x}, y)} = \frac{p_{\mathrm{te}}(\boldsymbol{z})}{p_{\mathrm{tr}}(\boldsymbol{z})} = w^*(\boldsymbol{z}). \tag{4}$$

*Proof.* Let $F_{\mathrm{tr}}(\boldsymbol{x}, y)$, $F_{\mathrm{te}}(\boldsymbol{x}, y)$, $F_{\mathrm{tr}}(\boldsymbol{z})$ as well as $F_{\mathrm{te}}(\boldsymbol{z})$ be the corresponding cumulative distribution functions (CDFs). By the definition of CDFs, the fundamental theorem of calculus,[2] and three properties of $\pi$ namely $\pi$ is fixed, deterministic and invertible, it holds that

$$p_{\mathrm{tr}}(\boldsymbol{x}, y)\mathrm{d}\boldsymbol{x} = \mathrm{d}F_{\mathrm{tr}}(\boldsymbol{x}, y) = \mathrm{d}F_{\mathrm{tr}}(\boldsymbol{z}) = p_{\mathrm{tr}}(\boldsymbol{z})\mathrm{d}\boldsymbol{z}, \tag{5}$$
$$p_{\mathrm{te}}(\boldsymbol{x}, y)\mathrm{d}\boldsymbol{x} = \mathrm{d}F_{\mathrm{te}}(\boldsymbol{x}, y) = \mathrm{d}F_{\mathrm{te}}(\boldsymbol{z}) = p_{\mathrm{te}}(\boldsymbol{z})\mathrm{d}\boldsymbol{z}, \tag{6}$$

where $\mathrm{d}$ denotes the differential operator, and

$$\mathrm{d}F_*(\boldsymbol{x}, y) = \tfrac{\partial}{\partial \boldsymbol{x}}\big(\sum_{y' \le y} \int_{\boldsymbol{x}' \le \boldsymbol{x}} p_*(\boldsymbol{x}', y')\mathrm{d}\boldsymbol{x}' - \sum_{y' < y} \int_{\boldsymbol{x}' \le \boldsymbol{x}} p_*(\boldsymbol{x}', y')\mathrm{d}\boldsymbol{x}'\big) \cdot \mathrm{d}\boldsymbol{x}.$$

For simplicity, the continuous random variable $\boldsymbol{x}$ and the discrete random variable $y$ are considered separately. Dividing Eq. (6) by Eq. (5) proves Eq. (4). □

Theorem 1 requires that $\pi$ satisfies three properties: we cannot guarantee $\mathrm{d}F_{\mathrm{tr}}(\boldsymbol{z}) = p_{\mathrm{tr}}(\boldsymbol{z})\mathrm{d}\boldsymbol{z}$ if $\pi$ is not fixed or $\mathrm{d}F_{\mathrm{tr}}(\boldsymbol{x}, y) = \mathrm{d}F_{\mathrm{tr}}(\boldsymbol{z})$ if $\pi$ is not deterministic or invertible. As a result, when $\mathcal{W}$ is computed in WE, $\boldsymbol{f}_\theta$ is regarded as fixed, and it could be switched to the *evaluation mode* from the

$$p_{\mathrm{tr}}(\boldsymbol{x}, y)\mathrm{d}|N_{\boldsymbol{x},y}| = \mathrm{d}\mu_{\boldsymbol{x},y,\mathrm{tr}}(N_{\boldsymbol{x},y}) = \mathrm{d}\mu_{\boldsymbol{z},\mathrm{tr}}(\pi(N_{\boldsymbol{x},y})) = p_{\mathrm{tr}}(\boldsymbol{z})\mathrm{d}|\pi(N_{\boldsymbol{x},y})|,$$

where $\mu_{\boldsymbol{x},y,\mathrm{tr}}$ and $\mu_{\boldsymbol{z},\mathrm{tr}}$ are the corresponding probability measures, $\pi(N_{\boldsymbol{x},y}) = \{\pi(\boldsymbol{x}', y') \mid (\boldsymbol{x}', y') \in N_{\boldsymbol{x},y}\}$, and $|\cdot|$ denotes the Lebesgue measure of a set. This more formal proof may be more than needed, since $w^*$ is estimable only if $p_*(\boldsymbol{x})$ are continuous and $F_*(\boldsymbol{x})$ are continuously differentiable.

*training mode* to avoid the randomness due to dropout [47] or similar randomized algorithms. The invertibility of $\pi$ is non-trivial: it assumes that $\mathcal{X} \times \mathcal{Y}$ is generated by a manifold $\mathcal{M} \subset \mathbb{R}^{d_{\mathrm{m}}}$ with an intrinsic dimension $d_{\mathrm{m}} \le d_{\mathrm{r}}$, and $\pi^{-1}$ recovers the generating function from $\mathcal{M}$ to $\mathcal{X} \times \mathcal{Y}$. If $\pi$ is from parts of $\boldsymbol{f}_\theta$, $\boldsymbol{f}_\theta$ must be a reasonably good classifier so that $\pi$ compresses $\mathcal{X} \times \mathcal{Y}$ back to $\mathcal{M}$. This finding is the circular dependency in Figure 1, which is the major theoretical contribution.

**Practical choices of $\pi$**  It seems obvious that $\pi$ can be $\boldsymbol{f}_\theta$ as a whole or without its topmost layer. However, the latter drops $y$ and corresponds to assuming

$$p_{\mathrm{tr}}(y \mid \boldsymbol{x}) = p_{\mathrm{te}}(y \mid \boldsymbol{x}) \implies \frac{p_{\mathrm{te}}(\boldsymbol{x},y)}{p_{\mathrm{tr}}(\boldsymbol{x},y)} = \frac{p_{\mathrm{te}}(\boldsymbol{x}) \cdot p_{\mathrm{te}}(y|\boldsymbol{x})}{p_{\mathrm{tr}}(\boldsymbol{x}) \cdot p_{\mathrm{tr}}(y|\boldsymbol{x})} = \frac{p_{\mathrm{te}}(\boldsymbol{x})}{p_{\mathrm{tr}}(\boldsymbol{x})} = \frac{p_{\mathrm{te}}(\boldsymbol{z})}{p_{\mathrm{tr}}(\boldsymbol{z})}, \tag{7}$$

which is only possible under *covariate shift* [38, 46, 50, 51]. It is conceptually a bad idea to attach $y$ to the latent representation of $\boldsymbol{x}$, since the distance metric on $\mathcal{Y}$ is completely different. A better idea to take the information of $y$ into account consists of three steps. First, estimate $p_{\mathrm{te}}(y)/p_{\mathrm{tr}}(y)$; second, partition $\{(\boldsymbol{x}_i^{\mathrm{tr}}, y_i^{\mathrm{tr}})\}_{i=1}^{n_{\mathrm{tr}}}$ and $\{(\boldsymbol{x}_i^{\mathrm{v}}, y_i^{\mathrm{v}})\}_{i=1}^{n_{\mathrm{v}}}$ according to $y$; third, invoke WE $k$ times on $k$ partitions separately based on the following identity: let $w_y^* = p_{\mathrm{te}}(y)/p_{\mathrm{tr}}(y)$, then

$$\frac{p_{\mathrm{te}}(\boldsymbol{x},y)}{p_{\mathrm{tr}}(\boldsymbol{x},y)} = \frac{p_{\mathrm{te}}(y) \cdot p_{\mathrm{te}}(\boldsymbol{x}|y)}{p_{\mathrm{tr}}(y) \cdot p_{\mathrm{tr}}(\boldsymbol{x}|y)} = w_y^* \cdot \frac{p_{\mathrm{te}}(\boldsymbol{x}|y)}{p_{\mathrm{tr}}(\boldsymbol{x}|y)} = w_y^* \cdot \frac{p_{\mathrm{te}}(\boldsymbol{z}|y)}{p_{\mathrm{tr}}(\boldsymbol{z}|y)}. \tag{8}$$

That being said, in a small mini-batch, invoking WE $k$ times on $k$ even smaller partitions might be remarkably unreliable than invoking it once on the whole mini-batch.

To this end, we propose an alternative choice $\pi : (\boldsymbol{x}, y) \mapsto \ell(\boldsymbol{f}_\theta(\boldsymbol{x}), y)$ that is motivated as follows. In practice, we are not sure about the existence of $\mathcal{M}$, we cannot check whether $d_{\mathrm{m}} \le d_{\mathrm{r}}$ when $\mathcal{M}$ indeed exists, or it is computationally hard to confirm that $\pi$ is invertible. Consequently, Eqs. (7-8) may not hold or only hold approximately. As a matter of fact, Eq. (1) also only hold approximately after replacing the expectations with empirical averages, and then it may be too much to stick with $w^*(\boldsymbol{x}, y)$. According to Eq. (1), there exists $w(\boldsymbol{x}, y)$ such that for all possible $f(\boldsymbol{x}, y)$,

$$\frac{1}{n_{\mathrm{v}}} \sum_{i=1}^{n_{\mathrm{v}}} f(\boldsymbol{x}_i^{\mathrm{v}}, y_i^{\mathrm{v}}) \approx \mathbb{E}_{p_{\mathrm{te}}(\boldsymbol{x},y)}[f(\boldsymbol{x}, y)] \approx \mathbb{E}_{p_{\mathrm{tr}}(\boldsymbol{x},y)}[w(\boldsymbol{x}, y) f(\boldsymbol{x}, y)] \approx \frac{1}{n_{\mathrm{tr}}} \sum_{i=1}^{n_{\mathrm{tr}}} w_i f(\boldsymbol{x}_i^{\mathrm{tr}}, y_i^{\mathrm{tr}}),$$

where $w_i = w(\boldsymbol{x}_i^{\mathrm{tr}}, y_i^{\mathrm{tr}})$ for $i = 1, \ldots, n_{\mathrm{tr}}$. This goal, *IW for everything*, is too general and its only solution is $w_i = w_i^*$; nonetheless, it is more than needed—*IW for classification* was the goal.

Specifically, the goal of DIW is to find a set of weights $\mathcal{W} = \{w_i\}_{i=1}^{n_{\mathrm{tr}}}$ such that for $\ell(\boldsymbol{f}_\theta(\boldsymbol{x}), y)$,

$$\frac{1}{n_{\mathrm{v}}} \sum_{i=1}^{n_{\mathrm{v}}} \ell(\boldsymbol{f}_\theta(\boldsymbol{x}_i^{\mathrm{v}}), y_i^{\mathrm{v}})\big|_{\theta=\theta_t} \approx \frac{1}{n_{\mathrm{tr}}} \sum_{i=1}^{n_{\mathrm{tr}}} w_i \ell(\boldsymbol{f}_\theta(\boldsymbol{x}_i^{\mathrm{tr}}), y_i^{\mathrm{tr}})\big|_{\theta=\theta_t}, \tag{9}$$

where the left- and right-hand sides are conditioned on $\theta = \theta_t$, and $\theta_t$ holds model parameters at a certain time point of training. After $\mathcal{W}$ is found, $\theta_t$ will be updated to $\theta_{t+1}$, and the current $\boldsymbol{f}_\theta$ will move to the next $\boldsymbol{f}_\theta$; then, we need to find a new set of weights satisfying Eq. (9) again. Compared with the general goal of IW, the goal of DIW is special and easy to achieve, and then there may be many different solutions, any of which can be used to replace $\mathcal{W}^* = \{w_i^*\}_{i=1}^{n_{\mathrm{tr}}}$ in $\widehat{R}(\boldsymbol{f}_\theta)$ in Eq. (3). The above argument elaborates the motivation of $\pi : (\boldsymbol{x}, y) \mapsto \ell(\boldsymbol{f}_\theta(\boldsymbol{x}), y)$. This is possible thanks to the *dynamic nature of weights* in DIW, which is the major methodological contribution.

**Distribution matching**  Finally, we perform distribution matching between the set of transformed training data $\{\boldsymbol{z}_i^{\mathrm{tr}}\}_{i=1}^{n_{\mathrm{tr}}}$ and the set of transformed validation data $\{\boldsymbol{z}_i^{\mathrm{v}}\}_{i=1}^{n_{\mathrm{v}}}$. Let $\mathcal{H}$ be a Hilbert space of real-valued functions on $\mathbb{R}^{d_{\mathrm{r}}}$ with an inner product $\langle \cdot, \cdot \rangle_{\mathcal{H}}$, or $\mathcal{H}$ be a *reproducing kernel Hilbert space*, where $k : (\boldsymbol{z}, \boldsymbol{z}') \mapsto \langle \phi(\boldsymbol{z}), \phi(\boldsymbol{z}') \rangle_{\mathcal{H}}$ is the reproducing kernel of $\mathcal{H}$ and $\phi : \mathbb{R}^{d_{\mathrm{r}}} \to \mathcal{H}$ is the kernel-induced feature map [44]. Then, we perform *kernel mean matching* [21] as follows.

Let $\mu_{\mathrm{tr}} = \mathbb{E}_{p_{\mathrm{tr}}(\boldsymbol{x},y) \cdot w(\boldsymbol{z})}[\phi(\boldsymbol{z})]$ and $\mu_{\mathrm{te}} = \mathbb{E}_{p_{\mathrm{te}}(\boldsymbol{x},y)}[\phi(\boldsymbol{z})]$ be the kernel embeddings of $p_{\mathrm{tr}} \cdot w$ and $p_{\mathrm{te}}$ in $\mathcal{H}$, then the *maximum mean discrepancy* (MMD) [3, 13] is defined as

$$\sup_{\|f\|_{\mathcal{H}} \le 1} \mathbb{E}_{p_{\mathrm{tr}}(\boldsymbol{x},y) \cdot w(\boldsymbol{z})}[f(\boldsymbol{z})] - \mathbb{E}_{p_{\mathrm{te}}(\boldsymbol{x},y)}[f(\boldsymbol{z})] = \|\mu_{\mathrm{tr}} - \mu_{\mathrm{te}}\|_{\mathcal{H}},$$

and the squared MMD can be approximated by

$$\|\mu_{\mathrm{tr}} - \mu_{\mathrm{te}}\|_{\mathcal{H}}^2 \approx \|\frac{1}{n_{\mathrm{tr}}} \sum_{i=1}^{n_{\mathrm{tr}}} w_i \phi(\boldsymbol{z}_i^{\mathrm{tr}}) - \frac{1}{n_{\mathrm{v}}} \sum_{i=1}^{n_{\mathrm{v}}} \phi(\boldsymbol{z}_i^{\mathrm{v}})\|_{\mathcal{H}}^2 \propto \boldsymbol{w}^\top \boldsymbol{K} \boldsymbol{w} - 2 \boldsymbol{k}^\top \boldsymbol{w} + \mathrm{Const.}, \tag{10}$$

where $\boldsymbol{w} \in \mathbb{R}^{n_{\mathrm{tr}}}$ is the weight vector, $\boldsymbol{K} \in \mathbb{R}^{n_{\mathrm{tr}} \times n_{\mathrm{tr}}}$ is a kernel matrix such that $\boldsymbol{K}_{ij} = k(\boldsymbol{z}_i^{\mathrm{tr}}, \boldsymbol{z}_j^{\mathrm{tr}})$, and $\boldsymbol{k} \in \mathbb{R}^{n_{\mathrm{tr}}}$ is a vector such that $\boldsymbol{k}_i = \frac{n_{\mathrm{tr}}}{n_{\mathrm{v}}} \sum_{j=1}^{n_{\mathrm{v}}} k(\boldsymbol{z}_i^{\mathrm{tr}}, \boldsymbol{z}_j^{\mathrm{v}})$. In practice, Eq. (10) is minimized subject to $0 \le w_i \le B$ and $|\frac{1}{n_{\mathrm{tr}}} \sum_{i=1}^{n_{\mathrm{tr}}} w_i - 1| \le \epsilon$ where $B > 0$ and $\epsilon > 0$ are hyperparameters as the upper bound of weights and the slack variable of $\frac{1}{n_{\mathrm{tr}}} \sum_{i=1}^{n_{\mathrm{tr}}} w_i = 1$. Eq. (10) is the objective of WE. The whole DIW is shown in Algorithm 1, which is our major algorithmic contribution.

**Algorithm 1** Dynamic importance weighting (in a mini-batch).

---

**Require:** a training mini-batch $\mathcal{S}^{\mathrm{tr}}$, a validation mini-batch $\mathcal{S}^{\mathrm{v}}$, the current model $\boldsymbol{f}_{\theta_t}$

**Hidden-layer-output transformation version:**

1: `forward` the input parts of $\mathcal{S}^{\mathrm{tr}}$ & $\mathcal{S}^{\mathrm{v}}$
2: `retrieve` the hidden-layer outputs $\mathcal{Z}^{\mathrm{tr}}$ & $\mathcal{Z}^{\mathrm{v}}$
3: `partition` $\mathcal{Z}^{\mathrm{tr}}$ & $\mathcal{Z}^{\mathrm{v}}$ into $\{\mathcal{Z}_y^{\mathrm{tr}}\}_{y=1}^k$ & $\{\mathcal{Z}_y^{\mathrm{v}}\}_{y=1}^k$
4: **for** $y = 1, \dots, k$ **do**
5:   `match` $\mathcal{Z}_y^{\mathrm{tr}}$ & $\mathcal{Z}_y^{\mathrm{v}}$ to obtain $\mathcal{W}_y$
6:   `multiply` all $w_i \in \mathcal{W}_y$ by $w_y^*$
7: **end for**
8: `compute` the loss values of $\mathcal{S}^{\mathrm{tr}}$ as $\mathcal{L}^{\mathrm{tr}}$
9: `weight` the empirical risk $\widehat{R}(\boldsymbol{f}_\theta)$ by $\{\mathcal{W}_y\}_{y=1}^k$
10: `backward` $\widehat{R}(\boldsymbol{f}_\theta)$ and `update` $\theta$

**Loss-value transformation version:**

1: `forward` the input parts of $\mathcal{S}^{\mathrm{tr}}$ & $\mathcal{S}^{\mathrm{v}}$
2: `compute` the loss values as $\mathcal{L}^{\mathrm{tr}}$ & $\mathcal{L}^{\mathrm{v}}$
3: `match` $\mathcal{L}^{\mathrm{tr}}$ & $\mathcal{L}^{\mathrm{v}}$ to obtain $\mathcal{W}$
4: `weight` the empirical risk $\widehat{R}(\boldsymbol{f}_\theta)$ by $\mathcal{W}$
5: `backward` $\widehat{R}(\boldsymbol{f}_\theta)$ and `update` $\theta$

---

# 3 Applications

We have proposed DIW for deep learning under distribution shift (DS). DS can be observed almost everywhere in the wild, for example, covariate shift, class-prior shift, and label noise.

*Covariate shift* may be the most popular DS, as defined in Eq. (7) [38, 46, 50, 51, 63]. It is harmful though $p(y \mid \boldsymbol{x})$ does not change, since the expressive power of $\boldsymbol{f}_\theta$ is limited, so that $\boldsymbol{f}_\theta$ will focus more on the regions where $p_{\mathrm{tr}}(\boldsymbol{x})$ is higher but not where $p_{\mathrm{te}}(\boldsymbol{x})$ is higher.

*Class-prior shift* may be the simplest DS, defined by plugging $p_{\mathrm{tr}}(\boldsymbol{x} \mid y) = p_{\mathrm{te}}(\boldsymbol{x} \mid y)$ in Eq. (8) so that only $p(y)$ changes [23, 17, 62, 20, 4, 29], whose optimal solution is $w^*(\boldsymbol{x}, y) = p_{\mathrm{te}}(y)/p_{\mathrm{tr}}(y)$, involving *counting* instead of *density ratio estimation* [52]. It is however very important, otherwise $\boldsymbol{f}_\theta$ will emphasize over-represented classes and neglect under-represented classes, which may raise transferability or fairness issues [6]. It can also serve as a unit test to see if an IW method is able to recover $w^*(\boldsymbol{x}, y)$ without being told that the shift is indeed class-prior shift.

*Label noise* may be the hardest or already adversarial DS where $p_{\mathrm{tr}}(\boldsymbol{x}) = p_{\mathrm{te}}(\boldsymbol{x})$ and $p_{\mathrm{tr}}(y \mid \boldsymbol{x}) \neq p_{\mathrm{te}}(y \mid \boldsymbol{x})$ which is opposite to covariate shift. There is a label corruption process $p(\tilde{y} \mid y, \boldsymbol{x})$ where $\tilde{y}$ denotes the corrupted label so that $p_{\mathrm{tr}}(\tilde{y} \mid \boldsymbol{x}) = \sum_y p(\tilde{y} \mid y, \boldsymbol{x}) \cdot p_{\mathrm{te}}(y \mid \boldsymbol{x})$, i.e., a label $y$ may flip to every corrupted label $\tilde{y} \neq y$ with a probability $p(\tilde{y} \mid y, \boldsymbol{x})$. It is extremely detrimental to training, since an over-parameterized $\boldsymbol{f}_\theta$ is able to fit any training data even with random labels [61]. Thus, label noise could significantly mislead $\boldsymbol{f}_\theta$ to fit $p_{\mathrm{tr}}(\tilde{y} \mid \boldsymbol{x})$ that is an improper map from $\boldsymbol{x}$ to $y$, and this is much more serious than misleading the focus of $\boldsymbol{f}_\theta$. Note that DIW can estimate $p(\tilde{y} \mid y, \boldsymbol{x})$, since our validation data carry the information about $p_{\mathrm{te}}(y \mid \boldsymbol{x})$; without validation data, $p(\tilde{y} \mid y, \boldsymbol{x})$ is unidentifiable, and it is usually assumed to be independent of $\boldsymbol{x}$ and simplified into $p(\tilde{y} \mid y)$, i.e., the *class-conditional noise* [37, 39, 30, 15, 14, 60, 55, 16, 58, 56, 59]. Besides label noise, DIW is applicable to similar DS where $p_{\mathrm{tr}}(\boldsymbol{x} \mid \tilde{y}) = \sum_y p(y \mid \tilde{y}) \cdot p_{\mathrm{te}}(\boldsymbol{x} \mid y)$ [45, 8, 34, 31, 32].

# 4 Discussions

Since DS is ubiquitous, many philosophies can handle it. In what follows, we discuss some related topics: learning to reweight, distributionally robust supervised learning, and domain adaptation.

*Learning to reweight* iterates between weighted classification on training data for updating $\boldsymbol{f}_\theta$, and unweighted classification on validation data for updating $\mathcal{W}$ [41]. Although it may look like IW, its philosophy is fairly different from IW: IW has a specific target $\mathcal{W}^*$ to estimate, while reweighting has a goal to optimize but no target to estimate; its goal is still empirical risk minimization on very limited validation data, and thus it may overfit the validation data. Technically, $\mathcal{W}$ is hidden in $\theta_{\mathcal{W}}$ in the objective of unweighted classification, so that [41] had to use a series of approximations just to differentiate the objective w.r.t. $\mathcal{W}$ through $\theta_{\mathcal{W}}$, which is notably more difficult than WE in DIW. This reweighting philosophy can also be used to train a *mentor network* for providing $\mathcal{W}$ [24].

*Distributionally robust supervised learning* (DRSL) assumes that there is no validation data drawn from $p_{\mathrm{te}}(\boldsymbol{x}, y)$ or $p_{\mathrm{te}}(\boldsymbol{x})$, and consequently its philosophy is to consider the worst-case DS within a

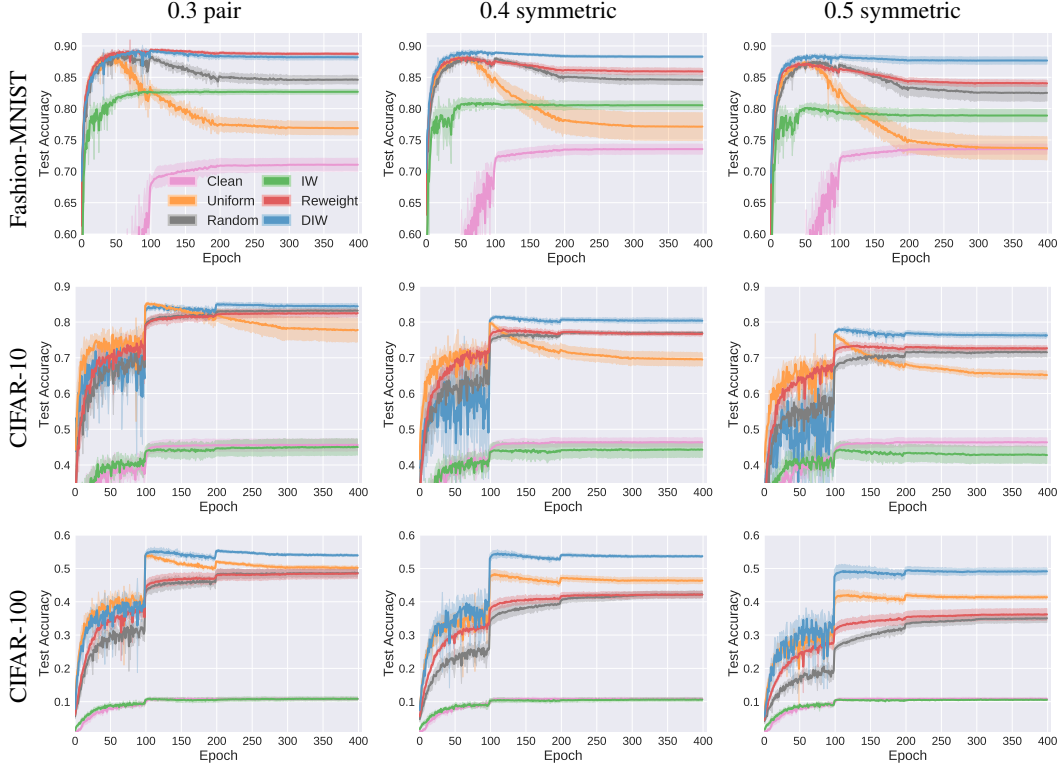

Figure 3: Experimental results on Fashion-MNIST and CIFAR-10/100 under label noise (5 trials).

prespecified uncertainty set [2, 54, 35, 36]. We can clearly see the difference: IW regards $p_{\text{te}}(\boldsymbol{x}, y)$ as fixed and $p_{\text{tr}}(\boldsymbol{x}, y)$ as shifted from $p_{\text{te}}(\boldsymbol{x}, y)$, while DRSL regards $p_{\text{tr}}(\boldsymbol{x}, y)$ as fixed and $p_{\text{te}}(\boldsymbol{x}, y)$ as shifted from $p_{\text{tr}}(\boldsymbol{x}, y)$. This worst-case philosophy makes DRSL more sensitive to bad training data (e.g., outliers or noisy labels) which results in less robust classifiers [19].

*Domain adaptation* (DA) is also closely related where $p_{\text{te}}(\boldsymbol{x}, y)$ and $p_{\text{tr}}(\boldsymbol{x}, y)$ are called in-domain and out-of-domain distributions [7] or called target and source domain distributions [1]. Although *supervised DA* is more similar to DIW, this area focuses more on *unsupervised DA* (UDA), i.e., the validation data come from $p_{\text{te}}(\boldsymbol{x})$ rather than $p_{\text{te}}(\boldsymbol{x}, y)$. UDA has at least three major philosophies: transfer knowledge from $p_{\text{tr}}(\boldsymbol{x})$ to $p_{\text{te}}(\boldsymbol{x})$ by bounding the *domain discrepancy* [10] or finding some *domain-invariant representations* [9], transfer from $p_{\text{tr}}(\boldsymbol{x} \mid y)$ to $p_{\text{te}}(\boldsymbol{x} \mid y)$ by conditional domain-invariant representations [11], and transfer from $p_{\text{tr}}(y \mid \boldsymbol{x})$ to $p_{\text{te}}(y \mid \boldsymbol{x})$ by pseudo-labeling target domain data [43]. They all have their own assumptions such as $p(y \mid \boldsymbol{x})$ or $p(\boldsymbol{x} \mid y)$ cannot change too much, and hence none of them can deal with the label-noise application of IW. Technically, the key difference of UDA from IW is that UDA methods do not weight/reweight source domain data.

## 5 Experiments

In this section, we verify the effectiveness of DIW.[3] We first compare it (loss-value transformation ver.) with baseline methods under label noise and class-prior shift. We then conduct many ablation studies to analyze the properties of SIW and DIW.

**Baselines** There are five baseline methods involved in our experiments:
- *Clean* discards the training data and uses the validation data for training;
- *Uniform* does not weight the training data, i.e., the weights are all ones;
- *Random* draws random weights following the *rectified Gaussian distribution*;
- *IW* is kernel mean matching without any non-linear transformation [21];
- *Reweight* is learning to reweight [41].

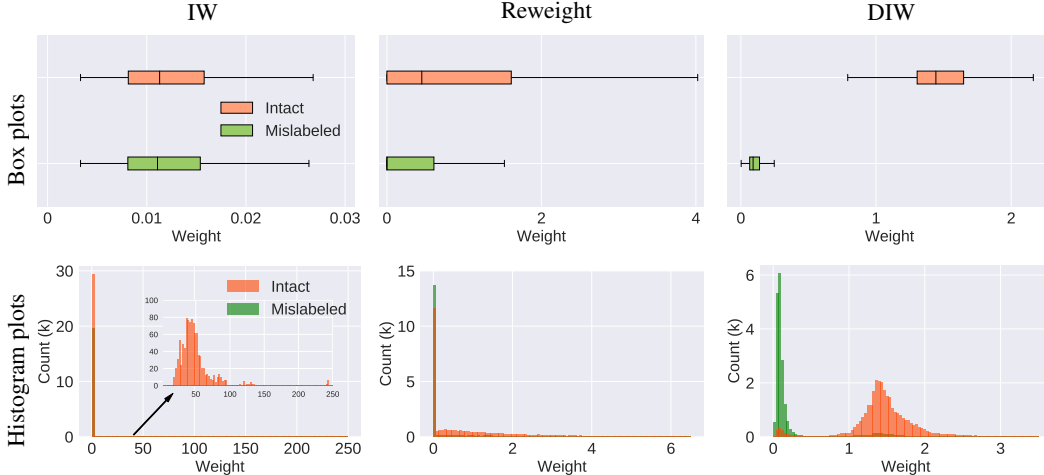

Figure 4: Statistics of weight distributions on CIFAR-10 under 0.4 symmetric flip.

All baselines are implemented with PyTorch.[4] Note that in each mini-batch, DIW computes $\mathcal{W}$ and then updates $\boldsymbol{f}_\theta$, while Reweight updates $\boldsymbol{f}_\theta$ and then updates $\mathcal{W}$. Moreover, Reweight updates $\mathcal{W}$ in epoch one, while DIW pretrains $\boldsymbol{f}_\theta$ in epoch one to equally go over all the training data once.

**Setup**   The experiments are based on three widely used benchmark datasets *Fashion-MNIST* [57], *CIFAR-10* and *CIFAR-100* [27]. For the set of validation data,

- 1,000 random clean data in total are used in the label-noise experiments;
- 10 random data per class are used in the class-prior-shift experiments.

The validation data are included in the training data, as required by Reweight. Then,

- for Fashion-MNIST, LeNet-5 [28] is trained by SGD [42];
- for CIFAR-10/100, ResNet-32 [18] is trained by Adam [26].

For fair comparisons, we normalize $\mathcal{W}$ to make $\frac{1}{n_{\text{tr}}} \sum_{i=1}^{n_{\text{tr}}} w_i = 1$ hold within each mini-batch. For clear comparisons, there is no data augmentation. More details can be found in the appendices.

**Label-noise experiments**   Two famous class-conditional noises are considered:

- *pair flip* [15], where a label $j$, if it gets mislabeled, must flip to class $(j \mod k + 1)$;
- *symmetric flip* [53], where a label may flip to all other classes with equal probability.

We set the noise rate as 0.3 for pair flip and 0.4 or 0.5 for symmetric flip. The experimental results are reported in Figure 3. We can see that DIW outperforms the baselines. As the noise rate increases, DIW stays reasonably robust and the baselines tend to overfit the noisy labels.

To better understand how DIW contributes to learning robust models, we take a look at the learned weights in the final epoch. As shown in Figure 4, DIW can successfully identify intact/mislabeled training data and automatically up-/down-weight them, while others cannot effectively identify them. This confirms that DIW can improve the weights and thus reduce the bias of the model.

**Class-prior-shift experiments**   We impose class-prior shift on Fashion-MNIST following [4]:

- the classes are divided into majority classes and minority classes, where the fraction of the minority classes is $\mu < 1$;
- the training data are drawn from every majority class using a sample size, and from every minority class using another sample size, where the ratio of these two sample sizes is $\rho > 1$;
- the test data are evenly sampled form all classes.

We fix $\mu = 0.2$ and try $\rho = 100$ and $\rho = 200$. A new baseline *Truth* is added for reference, where the true weights are used, i.e., $1 - \mu + \mu/\rho$ and $\mu + \rho - \mu\rho$ for the majority/minority classes.

The experimental results are reported in Table 1, where we can see that DIW again outperforms the baselines. Table 2 contains *mean absolute error* (MAE) and *root mean square error* (RMSE) from the weights learned by IW, Reweight and DIW to the true weights, as the unit test under class-prior shift. The results confirm that the weights learned by DIW are closer to the true weights.

Table 1: Mean accuracy (standard deviation) in percentage on Fashion-MNIST under class-prior shift (5 trials). Best and comparable methods (paired *t*-test at significance level 5%) are highlighted in bold.

|  | $\rho = 100$ | $\rho = 200$ |
|---|---|---|
| Clean | 63.38 (2.59) | 63.38 (2.59) |
| Uniform | **83.48 (1.26)** | 79.12 (1.18) |
| Random | **83.11 (1.70)** | 79.38 (0.96) |
| IW | **83.45 (1.10)** | **80.25 (2.23)** |
| Reweight | 81.96 (1.74) | **79.37 (2.38)** |
| DIW | **84.02 (1.82)** | **81.37 (0.95)** |
| Truth | **83.29 (1.11)** | **80.22 (2.13)** |

Table 2: Mean distance (standard deviation) from the learned weights to the true weights on Fashion-MNIST under class-prior shift (5 trials). Best and comparable methods (paired *t*-test at significance level 5%) are highlighted in bold.

| $\rho = 100$ | MAE | RMSE |
|---|---|---|
| IW | 1.10 (0.03) | 10.19 (0.33) |
| Reweight | 1.66 (0.02) | 5.65 (0.20) |
| DIW | **0.45 (0.02)** | **3.19 (0.07)** |

| $\rho = 200$ | MAE | RMSE |
|---|---|---|
| IW | 1.03 (0.04) | 9.99 (0.38) |
| Reweight | 1.64 (0.05) | 6.07 (0.86) |
| DIW | **0.46 (0.06)** | **3.67 (0.13)** |

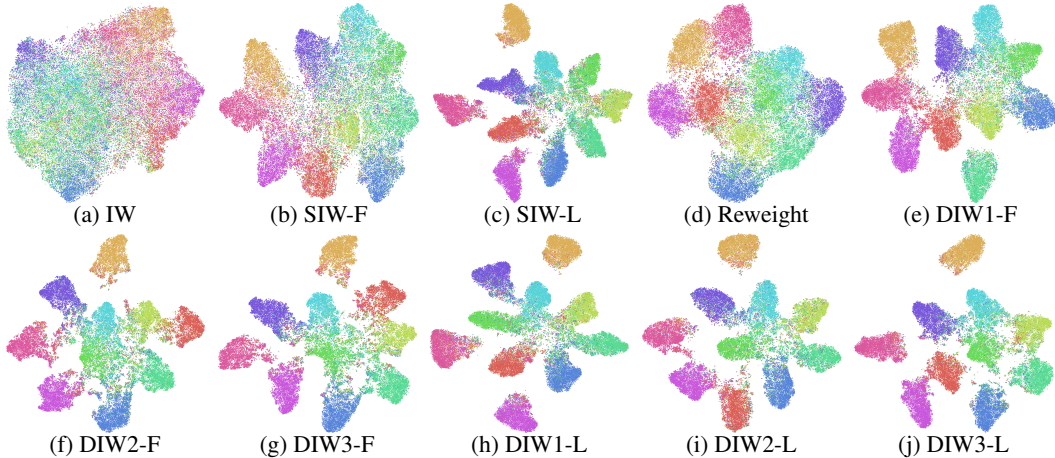

(a) IW  (b) SIW-F  (c) SIW-L  (d) Reweight  (e) DIW1-F

(f) DIW2-F  (g) DIW3-F  (h) DIW1-L  (i) DIW2-L  (j) DIW3-L

Figure 5: Visualizations of embedded data on noisy CIFAR-10 (colors mean ground-truth labels).

**Ablation study**    As shown in Figure 2, DIW comprises many options, which means that DIW can have a complicated algorithm design. Starting from IW,

- introducing feature extractor (FE) yields SIW;
- based on SIW, updating $\mathcal{W}$ yields DIW1;
- based on DIW1, updating FE yields DIW2;
- based on DIW2, pretraining FE yields DIW3.

We compare them under label noise and report the results in Table 3, where the "-F" or "-L" suffix means using the hidden-layer-output or loss-value transformation. In general, we can observe

- SIWs improve upon IW due to the introduction of FE;
- DIWs improve upon SIWs due to the dynamic nature of $\mathcal{W}$ in DIWs;
- for DIWs with a pretrained FE (i.e., DIW1 and DIW3), updating the FE during training is usually better than fixing it throughout training;
- for DIWs whose FE is updated (i.e., DIW2 and DIW3), "-F" methods perform better when FE is pretrained, while "-L" methods do not necessarily need to pretrain FE.

Therefore, DIW2-L is more recommended, which was indeed used in the previous experiments.

Furthermore, we train models on CIFAR-10 under 0.4 symmetric flip, project 64-dimensional last-layer representations of training data by *t-distributed stochastic neighbor embedding* (t-SNE) [33], and visualize the embedded data in Figure 5. We can see that DIWs have more concentrated clusters of the embedded data, which implies the superiority of DIWs over IW and SIWs.

Finally, we analyze the denoising effect of DIW2-L on CIFAR-10/100 in Figure 6 by the curves of the training accuracy on the intact data, mislabeled data (evaluated by the flipped and ground-truth labels) and the test accuracy. According to Figure 6, DIW2-L can simultaneously fit the intact data and denoise the mislabeled data, so that for the mislabeled data the flipped labels given for training correspond to much lower accuracy than the ground-truth labels withheld for training.

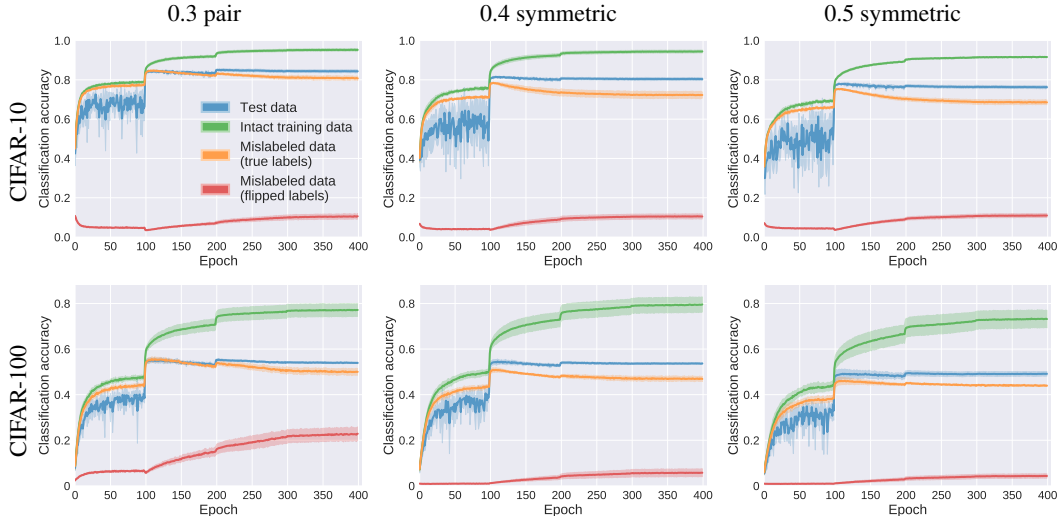

Figure 6: Denoising effect of DIW2-L on CIFAR-10/100 under label noise (5 trials).

Table 3: Mean accuracy (standard deviation) in percentage on Fashion-MNIST (F-MNIST for short) and CIFAR-10/100 under label noise (5 trials). Best and comparable methods (paired $t$-test at significance level 5%) are highlighted in bold. p/s is short for pair/symmetric flip.

| | Noise | IW | SIW-F | SIW-L | DIW1-F | DIW2-F | DIW3-F | DIW1-L | DIW2-L | DIW3-L |
|---|---|---|---|---|---|---|---|---|---|---|
| **F-MNIST** | 0.3 p | 82.69 (0.38) | 82.41 (0.46) | 85.46 (0.29) | 87.60 (0.07) | **87.67** **(0.37)** | 87.54 (0.25) | 87.04 (0.51) | **88.19** **(0.43)** | 86.68 (1.42) |
| | 0.4 s | 80.54 (0.66) | 82.36 (0.65) | **88.68** **(0.23)** | 87.45 (0.22) | 87.04 (0.30) | 88.29 (0.16) | **88.98** **(0.19)** | 88.29 (0.18) | 87.89 (0.43) |
| | 0.5 s | 78.90 (0.97) | 81.29 (0.68) | **87.49** **(0.23)** | **87.27** **(0.38)** | 86.41 (0.36) | 87.28 (0.18) | **87.70** **(0.15)** | **87.67** **(0.57)** | **86.74** **(1.19)** |
| **CIFAR-10** | 0.3 p | 45.02 (2.25) | 74.61 (0.51) | 80.45 (0.89) | 82.75 (0.57) | 81.19 (0.81) | 81.76 (0.70) | 81.73 (0.54) | **84.44** **(0.70)** | **83.80** **(0.93)** |
| | 0.4 s | 44.31 (2.14) | 65.58 (0.82) | 76.39 (0.72) | 78.23 (0.69) | 77.48 (0.60) | 78.75 (0.45) | 75.27 (1.37) | **80.40** **(0.69)** | **80.10** **(0.58)** |
| | 0.5 s | 42.84 (2.35) | 62.81 (1.29) | 71.47 (1.47) | 74.20 (0.81) | 73.98 (1.29) | **76.38** **(0.53)** | 69.67 (1.73) | **76.26** **(0.73)** | **76.86** **(0.44)** |
| **CIFAR-100*** | 0.3 p | 10.85 (0.59) | 10.44 (0.63) | 45.43 (0.71) | – | – | – | 51.90 (1.11) | **53.94** **(0.29)** | **54.01** **(0.93)** |
| | 0.4 s | 10.61 (0.53) | 11.70 (0.48) | 47.40 (0.34) | – | – | – | 50.99 (0.16) | **53.66** **(0.28)** | **53.07** **(0.32)** |
| | 0.5 s | 10.58 (0.17) | 13.26 (0.69) | 41.74 (1.68) | – | – | – | 46.25 (0.60) | **49.13** **(0.98)** | **49.11** **(0.90)** |

*Note that "-F" methods for DIW are not applicable on CIFAR-100, since there are too few data in a class in a mini-batch.

## 6 Conclusions

We rethought importance weighting for deep learning under distribution shift and explained that it suffers from a circular dependency conceptually and theoretically. To avoid the issue, we proposed DIW that iterates between weight estimation and weighted classification (i.e., deep classifier training), where features for weight estimation can be extracted as either hidden-layer outputs or loss values. Label-noise and class-prior-shift experiments demonstrated the effectiveness of DIW.

## 7 Broader impact

Distribution shift exists almost everywhere in the wild for reasons ranging from the subjective bias in data collection to the non-stationary environment. The shift poses threats for various applications of machine learning. For example, in the context of autonomous driving, the biased-to-training-data model may pose safety threats when applied in practice; and in a broader social science perspective, the selection bias in data preparation process may lead to fairness issues on gender, race or nation.

In this work, we aim to mitigate the distribution shift. We rethink the traditional importance weighting method in non-deep learning and propose a novel dynamic importance weighting framework that can leverage more expressive power of deep learning. We study it theoretically and algorithmically. As shown in the experiments, our proposed method can successfully learn robust classifiers under different forms of distribution shift. In ablation study, we also provide practical advices on algorithm design for practitioners.

**Acknowledgments**

We thank Prof. Magnus Boman and Prof. Henrik Boström for constructive suggestions in developing the work, and thank Xuyang Zhao, Tianyi Zhang, Yifan Zhang, Wenkai Xu, Feng Liu, Miao Xu and Ikko Yamane for helpful discussions. NL was supported by MEXT scholarship No. 171536 and the MSRA D-CORE Program. GN and MS were supported by JST AIP Acceleration Research Grant Number JPMJCR20U3, Japan.

## Footnotes

[1]After computing the new value of a weight, we discard its old value from the last epoch and only keep its new value. Instead, we can update a weight by convexly combining its old and new values. This may stabilize the weights across consecutive epochs, in case that WE is unstable when the batch size is very small.

[2]Here, it is implicitly assumed that PDFs $p_*(\boldsymbol{x})$ are Riemann-integrable and CDFs $F_*(\boldsymbol{x})$ are differentiable, and the proof is invalid if $p_*(\boldsymbol{x})$ are only Lebesgue-integrable and $F_*(\boldsymbol{x})$ are only absolutely continuous. The more formal proof is given as follows. Since $p_*(\boldsymbol{x}, y)$ are Lebesgue-Stieltjes-integrable, we can use probability measures: for example, let $N_{\boldsymbol{x},y} \ni (\boldsymbol{x}, y)$ be an arbitrary neighborhood around $(\boldsymbol{x}, y)$, then as $N_{\boldsymbol{x},y} \to (\boldsymbol{x}, y)$ where the convergence is w.r.t. the distance metric on $\mathcal{X} \times \mathcal{Y}$, it holds that

[3]Our implementation of DIW is available at `https://github.com/TongtongFANG/DIW`.

[4]We reimplement Reweight to ensure same random samplings of data and initialization of models.

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
