[Supplementary Material]

# Supplementary Material

## A  Supplementary information on experimental setup

In this section, we present supplementary information on experimental setup for label-noise and class-prior-shift experiments, and the implementation details for the methods discussed in ablation study. All experiments are implemented using PyTorch 1.6.0.

### A.1  Datasets and base models

**Fashion-MNIST**  Fashion-MNIST [57] is a 28*28 grayscale image dataset of fashion items in 10 classes. It contains 60,000 training images and 10,000 test images. See https://github.com/zalandoresearch/fashion-mnist for details.

The model for Fashion-MNIST is a LeNet-5 [28]:

|  |  |
|---|---|
| 0th (input) layer: | (32*32)- |
| 1st to 2nd layer: | C(5*5,6)-S(2*2)- |
| 3rd to 4th layer: | C(5*5,16)-S(2*2)- |
| 5th layer: | FC(120)- |
| 6th layer: | FC(84)-10 |

where C(5*5,6) means 6 channels of 5*5 convolutions followed by ReLU, S(2*2) means max-pooling layer with filter size 2*2 and stride 2, FC(120) means a fully connected layer with 120 outputs, etc.

**CIFAR-10 and CIFAR-100**  CIFAR-10 [27] is a collection of 60,000 real-world object images in 10 classes, 50,000 images for training and 10,000 for testing. Each class has 6,000 32*32 RGB images. CIFAR-100 [27] is just like the CIFAR-10, except it has a total number of 100 classes with 600 images in each class. See https://www.cs.toronto.edu/~kriz/cifar.html for details.

ResNet-32 [18] is used as the base model for CIFAR-10 and CIFAR-100:

|  |  |
|---|---|
| 0th (input) layer: | (32*32*3)- |
| 1st to 11th layers: | C(3*3, 16)-[C(3*3, 16), C(3*3, 16)]*5- |
| 12th to 21st layers: | [C(3*3, 32), C(3*3, 32)]*5- |
| 22nd to 31st layers: | [C(3*3, 64), C(3*3, 64)]*5- |
| 32nd layer: | Global Average Pooling-10/100 |

where the input is a 32*32 RGB image, [ ·, · ] means a building block [18] and [·]*2 means 2 such layers, etc. Batch normalization [22] is applied after convolutional layers. A dropout of 0.3 is added at the end of every building block.

### A.2  Label-noise experiments

The noisy labels are generated according to a predefined noise transition matrix $T$, where $T_{ij} = P(\tilde{y} = j | y = i)$. Two types of noise transition matrices are defined in Figure 7, where $\eta$ is the label-noise rate and $k$ is the number of classes. In pair flip label noise, a label $j$ may flip to class $(j \bmod k + 1)$ with probability $\eta$. In symmetric flip label noise, a label may flip to all other $k - 1$ classes with equal probability $\frac{\eta}{k-1}$. Note that the noise transition matrix and label-noise rate are unknown to the model.

$$
\begin{bmatrix}
1-\eta & \eta & 0 & \dots & 0 \\
0 & 1-\eta & \eta & \dots & 0 \\
\vdots & & \ddots & \ddots & \vdots \\
0 & 0 & \dots & 1-\eta & \eta \\
\eta & 0 & \dots & 0 & 1-\eta
\end{bmatrix}
\begin{bmatrix}
1-\eta & \frac{\eta}{k-1} & \dots & \frac{\eta}{k-1} & \frac{\eta}{k-1} \\
\frac{\eta}{k-1} & 1-\eta & \frac{\eta}{k-1} & \dots & \frac{\eta}{k-1} \\
\vdots & & \ddots & & \vdots \\
\frac{\eta}{k-1} & \dots & \frac{\eta}{k-1} & 1-\eta & \frac{\eta}{k-1} \\
\frac{\eta}{k-1} & \frac{\eta}{k-1} & \dots & \frac{\eta}{k-1} & 1-\eta
\end{bmatrix}
$$

Figure 7: Label-noise transition matrix. Left: Pair flip label noise; Right: Symmetric flip label noise.

For Fashion-MNIST experiments, SGD is used for optimization. The weight decay is 1e-4. For pair flip and symmetric flip, the initial learning rate is 0.0002 and 0.0003 respectively, decaying every 100 epochs by multiplying a factor of 0.1.

For CIFAR-10/100 experiments, Adam is used for optimization with its default parameters built in PyTorch 1.6.0. In CIFAR-10 experiments, the weight decay is 0.1 for pair flip and 0.05 for symmetric flip. For both pair and symmetric flip, the initial learning rate is 0.005, decaying every 100 epochs by multiplying a factor of 0.1. In CIFAR-100 experiments, the weight decay is 0.1 and the initial learning rate is 0.005, decaying every 100 epochs by multiplying a factor of 0.1 for both pair and symmetric flip.

For all label-noise experiments, the radial basis function (RBF) kernel is used in the distribution matching step: $k(\mathbf{z}, \mathbf{z}') = \exp(-\gamma \|\mathbf{z} - \mathbf{z}'\|^2)$, where $\gamma$ is 1-th quantile of the distances of training data. In the implementation, we use $\boldsymbol{K} + \omega I$ as the kernel matrix $\boldsymbol{K}$ in Eq 10, where $I$ is identity matrix and $\omega$ is set to be 1e-05. The upper bound of weights $B$ is 50 in Fashion-MNIST and 10 in CIFAR-10/100 experiments.

## A.3 Class-prior-shift experiments

To impose class-prior shift on Fashion-MNIST, we randomly select 10 data per class for validation set, 4,000 data (including the 10 validation data) per majority class for training set. The number of data per minority class (including the 10 validation data) in training set is computed according to $\rho$ as described in Section 5. We also randomly select 1,000 test data in class-prior-shift experiments. Majority class and minority class are randomly selected, where we use class 8 and 9 (i.e. Bag and Ankle boot) as the minority class and others (i.e. T-shirt/top, Trouser, Pullover, Dress, Coat, Sandal, Shirt and Sneaker) as majority class.

In class-prior-shift experiments, SGD is used for optimization. The weight decay is 1e-5 and the initial learning rate is 0.0005, decaying every epoch by multiplying a factor of 0.993. For the baseline "Clean" and "IW", the initial learning rate is 0.001 and 0.0003. Other hyperparameters are the same as other methods. Batch size is 256 for training and 100 for validation data. For the baseline "Truth", the ground-truth weights for majority class is calculated by:

$$w^*_{maj} = \frac{p_{\text{te}}(y)}{p_{\text{tr}}(y)} = \frac{1/k}{\rho n_s/(n_s \mu k + \rho n_s(1-\mu)k)} = 1 - \mu + \mu/\rho,$$

and for minority class is calculated by

$$w^*_{min} = \frac{p_{\text{te}}(y)}{p_{\text{tr}}(y)} = \frac{1/k}{n_s/(n_s \mu k + \rho n_s(1-\mu)k)} = \mu + \rho - \mu\rho,$$

where $k$ and $n_s$ are the number of total classes and the sample size of minority class respectively.

RBF kernel is again used in the distribution matching step, where $\gamma$ is 99-th quantile of the distances of training data. In the implementation, we use $\boldsymbol{K} + \omega I$ as the kernel matrix $\boldsymbol{K}$ in Eq 10, where $I$ is identity matrix and $\omega$ is set to be 1e-05. The upper bound of weights $B$ is 100.

## A.4 Methods in ablation study

We provide implementation details of the discussed methods in ablation study.

(1) IW:
- divide the training/validation data into $k$ partitions according to their given labels;
- perform weight estimation directly on the original data in each partition;
- perform weighted classification to train a DC using the learned static weights in the previous step, as shown in Figure 8a.

(2) SIW-F:
- divide the training/validation data into $k$ partitions according to their given labels;
- perform weight estimation on the hidden-layer-output transformations of data from a pretrained FE in each partition;
- perform weighted classification to train aother DC using the learned static weights in the previous step, as shown in Figure 8b.

(3) SIW-L:
- perform weight estimation on the loss-value transformations of data from a pretrained FE;
- perform weighted classification to train another DC using the learned static weights in the previous step, as shown in Figure 8b.
  (Note that "-L" methods do not need to partition data according to their given labels, because the label information is naturally included in the loss information.)

(4) DIW1-F:
- divide the training/validation data into $k$ partitions according to their given labels;
- for the current mini-batch, perform weight estimation on the hidden-layer-output transformations of data from a pretrained FE (in DC) in each partition;
- perform weighted classification to train another DC using the learned weights during training, and then move to the next mini-batch as shown in Figure 8c.

(5) DIW1-L:
- for the current mini-batch, perform weight estimation on the loss-value transformations of data from a pretrained FE (in DC);
- perform weighted classification to train another DC using the learned weights during training, and then move to the next mini-batch as shown in Figure 8c.
  (Note that for DIW1-F and DIW1-L, the FE is pretrained and fixed for weight estimation, and another DC is trained for weighted classification. But the learned weights are still dynamic due to the randomness of selected validation data in each mini-batch for performing weight estimation.)

(6) DIW2-F:
- divide the training/validation data into $k$ partitions according to their given labels;
- for the current mini-batch, perform weight estimation on the hidden-layer-output transformations of data from a randomly initialized FE (in DC) in each partition;
- perform weighted classification to train this DC using the learned weights during training, and then move to the next mini-batch as shown in Figure 8d.

(7) DIW2-L:
- for the current mini-batch, perform weight estimation on the loss-value transformations of data from a randomly initialized FE (in DC);
- perform weighted classification to train this DC using the learned weights during training, and then move to the next mini-batch as shown in Figure 8d.
  (Note that for DIW2-F and DIW2-L, the FE for weight estimation is in the same DC for weighted classification, so that they can be trained in a seamless manner.)

(8) DIW3-F:
- just like DIW2-F, except that the DC as FE is pretrained a little.

(9) DIW3-L:
- just like DIW2-L, except that the DC as FE is pretrained a little.

For all pretraining-based methods, we pretrain 20 epochs in Fashion-MNIST experiments and pretrain 50 epochs in CIFAR-10/100 experiments.

# B  Supplementary experimental results

In this section, we provide supplementary experimental results.

**Summary of classification accuracy**    Table 4 presents the mean accuracy and standard deviation on Fashion-MNIST, CIFAR-10 and CIFAR-100 under label noise. This table corresponds to Figure 3.

**Importance weights distribution on CIFAR-10**    Figure 9 shows the importance weights distribution on CIFAR-10 under 0.3 pair flip and 0.5 symmetric flip label noise, learned by DIW, reweight and IW. We can see that DIW can successfully identify intact/mislabeled training data and up-/down-weight them under different noise types.

Table 4: Mean accuracy (standard deviation) in percentage on Fashion-MNIST (F-MNIST for short), CIFAR-10/100 under label noise (5 trials). Best and comparable methods (paired *t*-test at significance level 5%) are highlighted in bold. p/s is short for pair/symmetric flip.

|  | Noise | Clean | Uniform | Random | IW | Reweight | DIW |
|---|---|---|---|---|---|---|---|
| F-MNIST | 0.3 p | 71.05 (1.03) | 76.89 (1.06) | 84.62 (0.68) | 82.69 (0.38) | **88.74 (0.19)** | 88.19 (0.43) |
|  | 0.4 s | 73.55 (0.80) | 77.13 (2.21) | 84.58 (0.76) | 80.54 (0.66) | 85.94 (0.51) | **88.29 (0.18)** |
|  | 0.5 s | 73.55 (0.80) | 73.70 (1.83) | 82.49 (1.29) | 78.90 (0.97) | 84.05 (0.51) | **87.67 (0.57)** |
| CIFAR-10 | 0.3 p | 45.62 (1.66) | 77.75 (3.27) | 83.20 (0.62) | 45.02 (2.25) | 82.44 (1.00) | **84.44 (0.70)** |
|  | 0.4 s | 45.61 (1.89) | 69.59 (1.83) | 76.90 (0.43) | 44.31 (2.14) | 76.69 (0.57) | **80.40 (0.69)** |
|  | 0.5 s | 46.35 (1.24) | 65.23 (1.11) | 71.56 (1.31) | 42.84 (2.35) | 72.62 (0.74) | **76.26 (0.73)** |
| CIFAR-100 | 0.3 p | 10.82 (0.44) | 50.20 (0.53) | 48.65 (1.16) | 10.85 (0.59) | 48.48 (1.52) | **53.94 (0.29)** |
|  | 0.4 s | 10.82 (0.44) | 46.34 (0.88) | 42.17 (1.05) | 10.61 (0.53) | 42.15 (0.96) | **53.66 (0.28)** |
|  | 0.5 s | 10.82 (0.44) | 41.35 (0.59) | 34.99 (1.19) | 10.58 (0.17) | 36.17 (1.74) | **49.13 (0.98)** |

(a) IW

(b) SIW

(c) DIW1

(d) DIW2 & DIW3

FE is short for feature extractor, LC/DC is for linear/deep classifier, and hid/loss stands for hidden-layer-output/loss-value transformation of data, denoting "-F"/"-L" method respectively. W is a set of weights. Circular update is employed to solve circular dependency.

Figure 8: Illustrations of IW, SIW and DIW.

Figure 9: Statistics of weight distributions on CIFAR-10 under 0.3 pair and 0.5 symmetric flips.