[Reviews · NeurIPS 2020]

Review 1

Summary and Contributions: This paper works on importance weighting for deep learning under distribution shift. Overall, it is of high quality. The idea is novel and the results are significant.

Strengths: The paper starts from the goal defined in equation 1. It finds out the bottleneck to apply existing algorithms is the circular dependency in figure 1. This bottleneck is also theoretically justified after theorem 1 in page 3. Then, the solution to the bottleneck is given in figure 2 (rough) and equations 8, 9, 10 and algorithm 1 (detailed). The equations and algorithm are clearly motivated. I think the solution is not only novel but also technically sound. Finally, the paper ends with applications and discussions (both like related work, closely and remotely).

Weaknesses: However, it is unclear how the experiments were done. Which version of SGD was used, and why not Adam or some better optimizer than Adam in deep learning? The authors claimed it is compatible with any model and any optimizer, but didn't show it is not limited to SGD. Moreover, why the convergence analysis is interesting? Can you show the assumptions hold even if the optimizer is limited to SGD? If so, how can you know the algorithm doesn't converge to a stupid local minimum which means the model or the algorithm itself is stupid? Last but not least, section 3 is fine but section 4 is a bit misleading. Importance weighting (including learning to reweight examples) seems more powerful, because you have labeled validation data, while unsupervised domain adaptation has unlabeled validation data and distributionally robust supervised learning even doesn't have unlabeled validation data. You should clearly tell this fact in this section. In summary, this is a nice paper with an important problem, a deep understanding, a novel idea, and a lot of experiments. Without enough information, I think the experiments are less convincing than they should be. The authors should also motivate their choices of model/optimizer, otherwise they may be cherry picked. The convergence analysis is confusing. I cannot get the message after carefully reading the paper. The merits outweigh the flaws and I vote to accept the paper. I would like to increase my score by one if the authors can convince me.

Correctness: yes

Clarity: yes

Relation to Prior Work: yes

Reproducibility: Yes

Additional Feedback:


Review 2

Summary and Contributions: This paper propose dynamic importance weighting as an end-to-end solution to distribution shift problem. They train a deep classifier on the importance-weighted training data as the feature extractor for importance weighting. The experimental results show that the proposed method exceed the traditional importance weighting methods.

Strengths: 1. IW is general and fundamental in machine learning against distribution shift. 2. The analysis is intuitive and insightful. 3. The proposed algorithm is novel and compatible with deep learning techniques. 4. The proposed algorithm experimentally outperforms learning to reweight.

Weaknesses: 1. In experiments of learning with noisy labels, the baseline algorithms are not latest. 2. Furthermore, the datasets are not enough. Should have a non-image dataset. 3. The distribution shift is synthetic. Should have a real-world dataset.

Correctness: Yes

Clarity: Yes

Relation to Prior Work: Yes

Reproducibility: Yes

Additional Feedback: ----------------------------- I have read the author's rebuttal, the rebuttal addressed the issue of latest baselines, and I decide to change my score to 8.


Review 3

Summary and Contributions: ===UPDATE AFTER AUTHOR FEEDBACK=== I think that the additional results and proposed experiments from the author feedback will really fill out the empirical contributions of this paper, and I have changed the overall score to reflect this. I think that this paper belongs in Neurips This paper introduces a method that simultaneously trains a deep network to perform (importance) weight estimation and classification using the same extracted/hidden features for a context involving both complex data and distribution shift where one desires to perform weighted ERM with a deep network. Their method DIW, alternates between updating the network based on a weight estimation objective and a weighted classification objective. The weight estimation objective is kernel mean matching on the distributions of either the final hidden layer representations or loss values between training data and a small validation set. They describe under which conditions the correct IWs for the raw data are also the correct IWs for the data under some transformation. They then motivate their choice for matching the loss, arguing that finding optimal weights to match data distributions is unnecessary. Finally, experiments are presented showing superior classification performance under label noise - results for the label shift scenario are also presented.

Strengths: The method is novel and interesting, and outperforms previous work empirically on image classification with label noise. Section 2 in the paper does a good job of justifying their choices in algorithm design. Since their final algorithm combines several design choices, I appreciate the inclusion of ablation studies that effectively illustrate how these individual choices affect model performance. They not only show superior classification results for the label noise scenario, but also show that their method is effective in downweighting mislabeled classes. Distribution shift is an important problem present in many real-world scenarios, and this paper is a step towards tackling the problem for complex modalities that require deep learning models.

Weaknesses: I don't understand the purpose of the class imbalance experiments. Weight estimation in label shift scenario only requires estimating the train/test ratio for each label, which is trivial with access to the labeled validation set. Weight estimation from extracted features seems like overkill here, and there is a glaring absence of a baseline that just uses the label ratios between train and validation set as the importance weights. Weighting for label shift can even be done effectively without validation set labels. There isn't any discussion of the results of this experiment at all in the paper either. The paper mentions that this experiment tests whether DIW can estimate weights without being told it's in the setting of class imbalance, but I think that experiments for the covariate shift scenario would be much more effective at demonstrating the weight estimation abilities of the proposed model, and it is also a harder task. (Additionally, I usually see the term "label shift" applied to this setting while "class imbalance" usually refers to one or more classes being more prevalent in the data regardless of distribution shift). For this type of deep learning methods paper, absent strong theoretical grounding, it's important to have thorough empirical results. While the label noise results presented are quite strong, the experiments only address one data modality (images) and don't address the covariate shift setting.

Correctness: Yes, I didn't see any flaws in correctness of claims and empirical methodology.

Clarity: The paper is very clear and well-written. The section that describes their method is easy to follow and well-organized.

Relation to Prior Work: Yes, the authors illustrate how their method differs from Learning to Reweight, as well as how their problem setup differs from similar ones in section 4.

Reproducibility: Yes

Additional Feedback: Experiments on text data and covariate shift, could make the empirical results very convincing. Additionally, I think it would be very interesting to look at how the weighted classifiers partition the train and test sets into different classes, i.e. what the ratios between between number of predicted and ground truth examples for different classes. It could be useful to briefly motivate the setting where you have access to a small labeled validation set drawn from the test distribution. This isn't the canonical supervised learning setup, so I think the paper should make an argument/motivation for the relevance of this scenario. I also think that this paper would benefit from having a real conclusion setting that contextualizes the method and results and examines the experimental results. While plotting accuracy against epochs in Figure 3, does illustrate the overfitting of other methods, it might be more interesting to plot accuracy vs amount of label noise to compare how the different methods perform across varying label flip rates. I's also like to note that whether or not performing weighted ERM with deep networks results in the desired effect can be heavily dependent on training time and hyperparameter choice, even when importance weights are known[Byrd, Lipton. What is the Effect of Importance Weighting in Deep Learning? ICML 2019], however you empirical results suggests that their method is still able to perform well.


Review 4

Summary and Contributions: ===UPDATE AFTER AUTHOR FEEDBACK=== Thanks to the authors for the clarifications. My score was not changed, I still believe that this is a good paper. Proposed experiments will make it stronger. The paper presents a new framework for training deep learning models in the presence of the distribution shift between train and test data. The proposed method iteratively optimises classifier for prediction task and estimates importance weights. Authors theoretically show the validity of the proposed approach, empirically compare their method with existing methods and perform ablation study.

Strengths: The proposed framework for dynamic estimation of importance weights is novel and potentially could be interesting for the community. The method is clearly described. Experiments are well designed and analysis of results highlights the main properties of the proposed approach.

Weaknesses: There are no experiments for 'covariate shift' type of distribution shift. To my opinion, the paper is incomplete without them and they are needed to support claims.

Correctness: Provided derivations and empirical methodology look correct.

Clarity: While the paper is clearly written, there is a couple of moments that can be improved: - It would be useful to add references for the first and second reason why it is difficult to boost the expressive power of WE (paragraph starting at line 38) - The paragraph starting at line 33 is a bit confusing and not really convincing. How the conclusion depends on the number of classes? Does w* use y as input or why does dimension equal to (d+1)?

Relation to Prior Work: There is a fairly good discussion section with a comparison of the proposed method and existing approaches. However, there is no explanation for a choice of baseline methods [20] and [38]. I would recommend highlighting the differences between them. For instance, explicitly mention that [20] doesn't require labels for a target test set to estimate IW.

Reproducibility: Yes

Additional Feedback:

[Author Response · NeurIPS 2020]

We thank all reviewers for the constructive comments! Following the suggestions, here is a list of new experiments.

(i) Added a baseline *Truth* for label shift where the importance weight is the validation/train ratio for each class:

| Acc | $\rho$ 100 | Truth: 83.05 (0.58) | DIW3-L: 83.69 (1.21) | $\rho$ 200 | Truth: 79.92 (0.46) | DIW3-L: 81.38 (1.24) |

Truth seems slightly worse since it is just roughly tuned; the difference is *statistically insignificant* by $t$-test.

(ii) Computed the $\ell_2$ distances between the true weights (by Truth) and the estimated weights (by other methods):

| Dist | $\rho = 200$ | IW: 0.0324 (0.0010) | Reweight: 0.0321 (0.0010) | DIW3-L: 0.0166 (0.0003) |

The weights by DIW3-L are *statistically significantly closer* to the true weights. It is a unit test possible under
label shift, which is why we wanted the label shift experiments, but we didn't finish them before the deadline.

(iii) Tested Adam as the optimizer (we can see that the messages *do not change much* if changing SGD to Adam):

| (CIFAR-10, 40% symmetric noise) Acc | IW: 44.51 (1.55) | Reweight: 72.96 (0.97) | DIW3-L: 79.80 (0.25) |

(iv) TODO: add text-data experiments (based on *20Newsgroups*) and real-world experiments (based on *Clothing1M*).
For 20Newsgroups, we need to choose the network architecture, which is not a trivial task. For Clothing1M, we
are using the common practice ResNet-50, but the issue is that it goes in a speed of 2 days/epoch...

(v) TODO: add covariate shift experiments based on MNIST, Fashion-MNIST, or CIFAR-10. It is quite difficult to
simulate covariate shifts based on benchmark datasets where we *cannot access/manipulate* $p(x)$ or $p(x|y)$, even
though covariate shift should be extremely popular in the wild. Most of covariate shift papers made use of fully
controlled toy data where *IW suffices* and there is *no circular dependency*.
We have two plans: first, we may try strongly regularized GANs (e.g., very early stopped); second, we may try
mixup style data generation (mixing two images taken from two classes). We are thinking of ways to guarantee
that the shift is precisely covariate shift and the shift is challenging to learning methods, i.e., $p(x)$ should change
significantly, $p(x)$ must have a larger support in training, $p(y|x)$ cannot change at all. Any advice is welcome.
In the next version, we will definitely include all the aforementioned experiments!

**Use of validation data and motivation of the problem setting** (by R2 & R4 & R5) Our setting is the general IW,
with the same data generation as *Learning to reweight* [38] and *MentorNet* [22] (but a different goal from them).
First of all, note that there is no *distributional assumption* about the *joint distribution shift*. Given that $p(x, y)$ can
shift now, without any further information, the shift is *not identifiable*, or equivalently, the weights are *not estimable*.
Hence, we should let the learning methods see some *clean validation data* coming from the test distribution. By careful
algorithm design, the learning methods should be able to infer the shift by referring to the clean validation data.
On the other hand, what was done in more specific shifts (i.e., covariate shift, label/class-prior shift, and label noise)
is that a specific distributional assumption is added instead of the clean validation data to the algorithm design:
• under covariate shift, only $p(x)$ can shift, and $p(y|x)$ must remain the same;
• under label shift, only $p(y)$ can shift, and $p(x|y)$ must remain the same;
• under label noise, only $p(y|x)$ can shift, and $p(x)$ must remain the same.
The learning methods should be able to infer *the parameters but not the type of* the shift, because the type is already
given by the assumption. In this sense, *IW is more general* than these specific shifts without clean validation data; the
*data-driven philosophy/methodology* can tackle more complex shifts where such an assumption is not obvious.
Last but not least, the small set of clean validation data can be obtained with the help of domain experts. Since it is
small, it would not cost too much of the data-labeling budget. Some real-world label-noise datasets are designed in this
manner, for example, Clothing1M whose clean training data can serve as our clean validation data.

**Baselines are not latest for learning with noisy labels** (by R3) The problem setting of the proposal is IW rather than
label noise that is only a special case of IW. The latest label-noise methods *may not be applied* in the more general IW.
Among the learning methods under the same problem setting (i.e., how the data look like), *[38] is already the latest*.

**Class imbalance to label shift** (by R4) Thanks for pointing the fact out! Yes, class imbalance refers to *cost-sensitive*
*learning* under *no label shift*, but the goal is to maximize the AUC, which has been shown as same as to minimize the
*balanced error* [Menon et al., ICML'15]. In our experiments, the test data is balanced so that the classification error
is also the balanced error. That being said, our terminology following [5] is not as good as label shift. We will call it
class-prior or *prior probability shift* following the book [37], since label shift may also be confused with label noise.

**On training time and hyperparameter choice** (by R4) Yes, we have used many tricks including regularizations and
learning rate decays. In fact, not only weighted ERM with IW but also standard ERM relies on the training time and
hyperparameter choice. Our contribution was to find that *complex data form is not necessarily the bottleneck of IW*.

**No experiments for the hidden-layer-output transformation version in Algorithm 1** (by R5) In our experiments,
*this version corresponds to "-F" methods*, and the other version corresponds to "-L" methods. Indeed, the experimental
results you wanted are reported and discussed in Section 5.2 (more specifically, in Table 2 and Figure 5).

[Meta-Review · NeurIPS 2020]

This paper works on importance weighting for deep learning under distribution shift. All reviewers agree that the manuscript is of high quality, that the ideas are novel and the results are significant.